# Preservation and Restoration of an Old Wooden Icon with Complex Carved Ornaments, in a Conservation State of Precollapse

**Liliana Nica [1], Viorica Vasilache [2,\*], Ana Drob [2,\*], Silvea Pruteanu [1] and Ion Sandu [2,3,4,5,\*]**

[1] Doctoral School of Geosciences, Faculty of Geography and Geology, "Alexandru Ioan Cuza" University of Iasi, Blvd. Carol I No. 11, 700506 Iasi, Romania; liliana_nica@yahoo.com (L.N.); silvia.lazarpruteanu@gmail.com (S.P.)

[2] Arheoinvest Center, Institute of Interdisciplinary Research, Department of Natural and Exact Sciences, "Alexandru Ioan Cuza" University of Iasi, Blvd. Carol I No. 11, 700505 Iasi, Romania

[3] Academy of Romanian Scientists AOSR, 54 Splaiul Independentei St., Sect. 5, 050094 Bucharest, Romania

[4] National Institute for Research and Development in Environmental Protection, 294 Splaiul Independentei, Sect. 6, 060031 Bucharest, Romania

[5] Romanian Inventors Forum, Sf. Petru Movila St., L11, 3-3, 700089 Iasi, Romania

\* Correspondence: viorica.vasilache@uaic.ro (V.V.); ana.drob@uaic.ro (A.D.); ion.sandu@uaic.ro (I.S.)

**Abstract:** Wooden icons used in liturgical activities suffer a series of evolutionary deteriorations and degradations over time, due to improper storage and use conditions. The deterioration of the physical state of the structural-functional elements and degradation of the chemical nature of the components often lead old easel paintings to precarious preservation (almost close to pre-collapse), impossible to use or display as an artifact. In this study was included an old oil-painted icon on a carved linden wood support with fine gilded ornaments, which frames a central icon ("Coronation of Virgin Mary") and a complex Menaion icon system with iconographic scenes. It was made by an anonymous author and dates back to the late 18th and early 19th centuries. As the icon has a special beauty and an ornamental and iconographic complexity, having a great heritage value, it required the elaboration of an optimal preservation–restoration protocol for the museum exhibition. Initially, the nature of the pictorial materials was determined, and their preservation state was evaluated using the OM, SEM-EDX, micro-FTIR methods, and CIE L\*a\*b\* colorimetry and visible and UV reflectography were used in the evaluation of the wash test and in compatibility studies. Based on the data obtained, the optimal materials and procedures for structural reintegration were selected (including support fillings and filling of gaps), then chromatic reintegration and gilding, followed by final revarnishing, with or without patination additives.

**Keywords:** oil-painted icon; carved wood stand; gilded ornaments; deterioration; degradation; preservation; restoration

## 1. Introduction

An old icon represents, at the time it was made, a painting with a liturgical message, which over the centuries becomes an independent entity, with certain patrimonial elements and functions. Its component materials and structural elements, exposed to environmental factors, improper storage and use conditions, often suffer from deterioration and evolutionary degradation, which affects its aesthetics and integrity, and thus the painter's message. Therefore, ensuring an optimal microclimate and compliance with scientific conservation norms is a permanent necessity, both for public and private collections [1–7].

Wooden icons painted in tempera or oil, whether varnished or not, often suffer over time due to the microclimate factors, careless handling and display, a series of micro- or macrostructural destruction processes and chemical, microbiological and thermal alterations in the substrate, preparation coatings, color layer/gild and varnish. For example, in

the case of very old pre-collapsed icons, the highly damaged protective layers (varnishes) are found with dirt deposits that lead to opaquisation and browning of the iconographic system (painting/gilding) by oxidative fouling and/or thermal/photochemical cornification. The paint layer, including the preparations and the panel, undergoes chromatic changes, embrittlement due to the loss of cohesive capacity of the binders, accidental carbonization, the substrate rottenness and other damages and degradations [8–12].

Icons with perforated supports and carved in complex profiles or with gilded ornamental frames interact strongly and differently with environmental factors, compared to those painted on a compact panel. *Deterioration* evolves differently at the level of each structural element, and *degradation*, depending on the nature of the material used in the installation. Thus, the support, the preparation, the color layer (polychromy) and the varnish, will interact with the environmental factors and agencies differently from the attached decorative elements (frames or carved ornaments), decorated by painting, gilding or metal foil plating.

If the deteriorations through *destruction processes* at the micro- or macroscopic level changes the physical state of the structural-functional elements, the *degradations* modify by *alteration* in the *chemical nature* of the component materials. For this reason, the *deterioration* will be related to the structural and/or functional element, and the *degradation* will be correlated with each type of material in the painting layer, gild and support [8,9,13–17].

In the case of *wood*, attention shall be paid to the essence (species), age of the tree and wood age (including the time that has elapsed since the tree was cut down) area and period of harvest, the method of processing, the insecticide treatments and the nature and application method of the protective layer of the reverse. For the *preparation*, the binder and the filler material are analyzed, and for the *polychrome layer* the pigments and the binder of the colors used. Instead, on the *varnish* of the painted surface and on the gildings, the nature of the materials and the application method are studied.

When using wood as a support for icons, it requires, after water stabilization, an insecto-fungal treatment, before or after mechanical processing, followed by specific preparations, in the form of a multifunctional layer with different thicknesses, called *primer* for painting and *bolus* for gilding, which often work differently under the influence of environmental factors [2,5,8,18,19].

Over time, dirt deposits, as a result of photooxidation and thermal processes, lead from opaquisation and yellowing of the varnish to its partial or total blackening when the chemical modification of the painting materials takes place, being identified as the appearance of mobile cracks, "blind" detachments, alveoli, superficial or deep gaps, etc. Such processes also occur in gilded ornaments, which are often severely affected by improper cleaning or regilding. Corrosive gases, dust and smoke emitted during liturgical rituals, along with direct display, without the use of protective systems (boxing or framing under glass) and careless handling, often lead to irreversible deterioration and degradation. Another important cause, in addition to the tar deposits resulting from the burning of candles, incense and candle oils, is the large fluctuations in temperature and humidity [2,6,18,20–22].

During the display, the deposits move from the *non-adherent* phase (easy to remove) to the *semi-adherent* phase and then *strongly adherent*, in the last phase the dirt interacts strongly with the varnish, the color or gilding layers. As a result of this, removing adherent dirt is a daunting task for restorers. Thus, depending on the nature, the state of conservation and the porosity of the pictorial materials, respectively the degree of dirt penetration, it is necessary to carry out a *cleaning protocol* based on the *washing tests*. Cleaning systems and processes must not affect the old patina, varnish, washers and slightly degraded ultra-thin color layers and have a minimal impact on the operator and the environment. They must be compatible and have a synergistic effect [23–39].

The stability and integrity of the pictorial layer, composed of varnish, color layer, drawing and preparation, are the main elements of the image and the artistic message [40–51]. For these reasons, before the cleaning operation, it is necessary to preventively strengthen the paint layer and the wooden support (in case of dynamic detachments), with Japanese

paper (veil)/fish glue, followed, if necessary, by preventive insecticide. After cleaning, the final consolidation and structural reintegration are performed by filling the missing elements, then filling the gaps, chromatic reintegration and, respectively, regilding, as finally it will be re-varnished by thin-layer with a transparent and reversible protective varnish [27–30]. When repatination is required, the varnish is lightly pigmented with burnt umber to restore the old patina. After applying the missing preparation, by grouting or priming, the repainting is performed by the techniques of reintegration/regilding and revarnishing. All preservation and restoration interventions will be limited to the affected areas, taking into account their depth, texture and area (thus respecting the principle of minimum intervention). The preparation of the painting layer structure must be obtained from a compatible binder and an inert filler material, such as: chalk dust or plaster, and for gilding it will be made of red pigment or a colored earth (bolus). They will be applied in successive layers, and the material used must be resistant to aging, elastic and flexible, with a texture similar to the old one, and be compatible with the original primer or bolus (respecting the principles of stability and compatibility of interventions) [2,18,31–33,52]. In the case of movable icons, used in external liturgical rituals, structural and chromatic reintegration are essential, as the rate of deterioration and degradation is very high, being often handled carelessly and without weather protection measures (used both in rain, hail, snow, fog, and sunny weather). Dimensional changes of the support under the influence of hygroscopic humidity and ambient temperature increase the risk of deteriorations caused by shrinkage and expansion [2,23]. Furthermore, being an organic material, the wooden support is continuously subjected to insecto-fungal attack, suffering evolutionary effects of biodeterioration and biodegradation, which is found in most of the old icons on wood, displayed or used improperly [2,8,9,31–33].

This paper focuses on the development of an interdisciplinary protocol based on the precarious conservation state of the artifact, for two research directions: the selection of new materials, and the preservation and restoration compatible procedures, in order to remove the artifact from the state of precollapse and to restore it, as much as is possible, to its original shape, which would allow museum display. Therefore, this study aims to preserve and restore an old heritage icon in precollapse (no longer allows display), using an innovative experimental protocol that includes a series of compatible materials and specific intervention operations, such as: preventive consolidation and stopping insecto-fungal attack, partial cleaning and varnishing, final completion and consolidation of the pictorial support and layers, chromatic reintegration of the paintings and partial regilding of the lacunar areas, followed by revarnishing with the restoration of the old patina.

The methods used to determine the chemical nature of the composite materials and to determine their state of preservation are OM, SEM-EDX, micro-FTIR, and CIE L*a*b* colorimetry and visible and UV reflectography were used in the evaluation of the wash test and in compatibility and synergy studies. Based on the data obtained, the optimal materials and procedures for structural reintegration were selected (including support fillings and filling of gaps), then chromatic reintegration and gilding, followed by final revarnishing, with or without patination additives.

## 2. Materials and Methods

### 2.1. Experimental Part

Icon Description and State of Conservation

The icon "Coronation of Virgin Mary" (Figure 1a) comes from a private collection and was painted in oil, in neoclassical style, in the late 18th century, on a perforated and carved linden wood support, with the panel consisting of a single countertop, without beams, with dimensions of $600 \times 445 \times 20$ mm. The edges of the wooden panel are attached to a thin wooden frame, briefly carved by incision with geomorphic elements. On the lower and upper edge, a place for fixing into a protection box was made by cutting into the central area (Figure 1b), which in time was lost. The existence of a protective box since it was made, demonstrated the artistic and spiritual value of this "family icon", having the same

"prestige" with the royal icons from a worship iconostasis. It has superficial and deep gaps in the proportion of 60%.

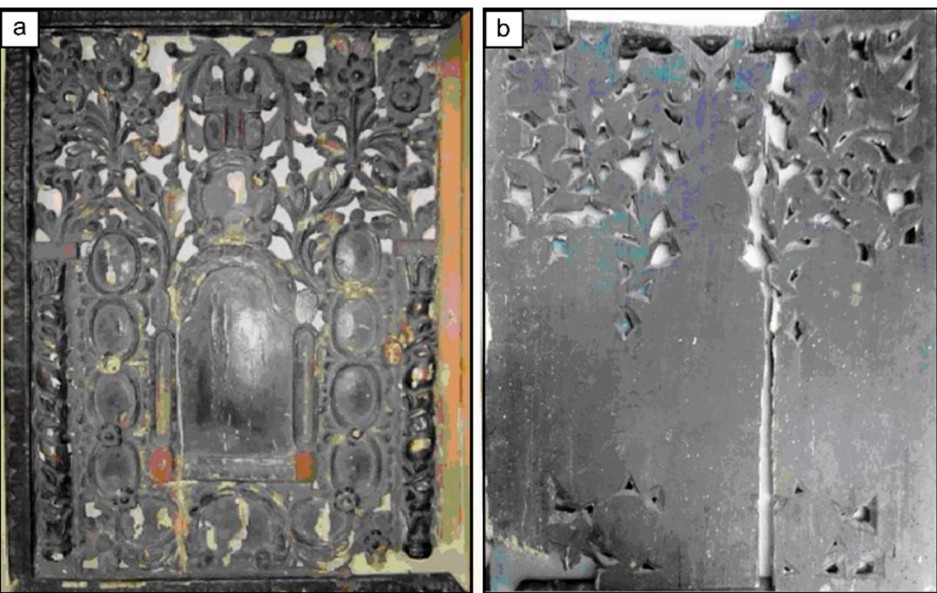

**Figure 1.** The icon "Coronation of Virgin Mary" in the initial stage of the study: (**a**) Front; (**b**) Back.

During the manufacturing, the water-stabilized linden wood support was perforated and carved with vegetal motifs (leaves and flowers braided in a harmonious symmetry), on which the preparation (primer) was applied on the entire surface, and then a bolus was brushed on the areas that were gilded with gold and silver foils. The sculptural ensemble integrates in harmony, with a high symmetry, a complex iconographic system, consisting of a central icon ("Coronation of the Virgin Mary"), framed laterally by two Menaion icons, with a carved frame, arched upwards and straight at the base, having at the top a "Crucifix" with two miniature icons on either side, placed on a medallion with the "Jesus Christ Mandylion", and at the bottom the scene is a representation of the "Last Supper". On either side of the central iconographic system are two series of five medallions, with the main saints and angels of Orthodoxy, interconnected by gilded flowers. Furthermore, the other ornaments are united by a richness of carved and gilded with gold and silver foil floral elements, between two twisted columns with carved leaves. From the top of these columns, of the pillars with plinths (base of the column-stylobate) and capitals (simple abacus) type, start the two stems with abundant floral elements, having symmetry towards the center. The 32 gilded floral elements have a silver gilded button in the center, and in the upper register, around the crucifix, there are another eight silver buttons.

The polychrome execution technique is classic and consists of a layer of primer applied on the finished wood carvings and medallions, without marouflage canvas. All the ornaments have a layer of ocher-red bolus over the primer, after which the thin metal sheets of gold and silver were glued (only a few flowers around the crucifix are covered with silver foil, the rest being with gold). The medallions are painted with colored earth-type pigments, mixed with boiled flax oil.

The icon is of a rare beauty through the symmetrical ornamental system, the artistic value being given by its complex filigree structure and miniature painting. The entire wooden panel has a rich and detailed sculpture with phytomorphic elements, which centrally frames the main icon, with the 12 medallions painted with different Saints.

The chisel marks on the back of the panel indicate that it was composed of a single countertop, which in time cracked longitudinally, splitting the icon in two.

The flowers, the buttons, the bases and capitals of the two columns, together with the frame rods, were applied by soldering animal glue on the high–flat areas of the panel. As a result of this, most of these elements and one of the frame rods, fell by detachment.

Furthermore, part of the leaves and the floral stem, visible in Figure 1a, in the upper half, but also in the lower right corner (slightly visible in Figure 1b on the left), were also lost as a result of the xylophagous attack. The presence of superficial restoration interventions is visible by covering all the gilded and silver-plated sculptures with bronze and the addition of wooden plates for support. Unfortunately, they were nailed down and caused much damage, which evolved over time. Following the previous restoration interventions, all the gilded surfaces were covered with bronze, and the haloes of the Saints were painted with yellow.

When taken over in the restoration lab, the state of preservation of the icon was close to precollapse, the support being strongly weakened due to the attack of xylophagous insects and fractured longitudinally in the form of two separate pieces. These two fragments showed several deep cracks, due to the curvature phenomenon, which greatly affected the painting of the central icon, but also to some of the medallions.

The custom of having the icon sanitized with lampante or vegetable oil, which over time has amplified the blackening of the polychrome, also contributed to the destruction of the icon. It completely affected the legibility of the icon, but also the aesthetics of the ornate support.

Previous restoration work by repainting the gilded areas is older than 100–110 years, as the dirt layer on it had the same thickness and uniformity as on the entire surface.

The photooxidation processes at the interface of the varnish-soot dirt interface led, by fouling, to the complete darkening of the iconographic image, and the extended xylophagous attack contributed to the appearance of holes and channels, which together with cracks, alveoli and crevices led to the loss of pictorial material, but also from the support, especially for fine gilded ornaments.

The central painting and the medallion ones showed roofing degradations and alveolation, which by breaking led to the loss of pictorial material on large areas. The damage to the central medallion was due to the wooden support being strongly weakened by the xylophagous attack and the processes of contraction and expansion, and the fracture to the left of the central icon (Figure 1b) became total after careless display and handling.

Initially, the icon, along with the lost boxing system, presented on the four sides framing rods, with floral ornaments carved superficially and gilded with gold, which also had a fixing role in the box.

Figure 2 shows some evolutionary damage to the structural-functional elements, along with the most representative degradations of the component materials.

Therefore, the conservation state of the icon required interventions for the preventive (reversible) consolidation of all elements in accordance with the experimental protocol.

### 2.2. Substantiation and Elaboration of the Experimental Protocol

In the elaboration of the protocol, the following steps were taken into account: multi-analytical scientific investigation to determine the nature of the materials and their state of preservation, application of preliminary consolidation and preventive insecto-fungal operations, wet cleaning of adherent deposits involving washing tests, final consolidation, grouting, chromatic reintegration by repainting and regilding, followed by revarnishing with or without repatination additives.

Particular attention was paid to the selection of analysis methods in the system of co-assistance and corroboration between interdisciplinary techniques, as well as new materials and compatible application procedures.

The first objective, that of multi-analytical scientific investigation, focused on three aspects:

- determining the nature of the materials used in production and assessing their state of conservation (highlighting the damage, respectively, static and dynamic/evolutionary degradations);
- establishing the nature of the dirt, the degree of penetration/interaction with the painting layer and the extent of the deposits, including the subsequent unauthorized repainting and regilding interventions;

- the use of modern non-invasive methods to obtain information with multiple uses in the process of museum display of the icon, and in particular, in assessing the effectiveness of cleaning operations and in compatibility studies;

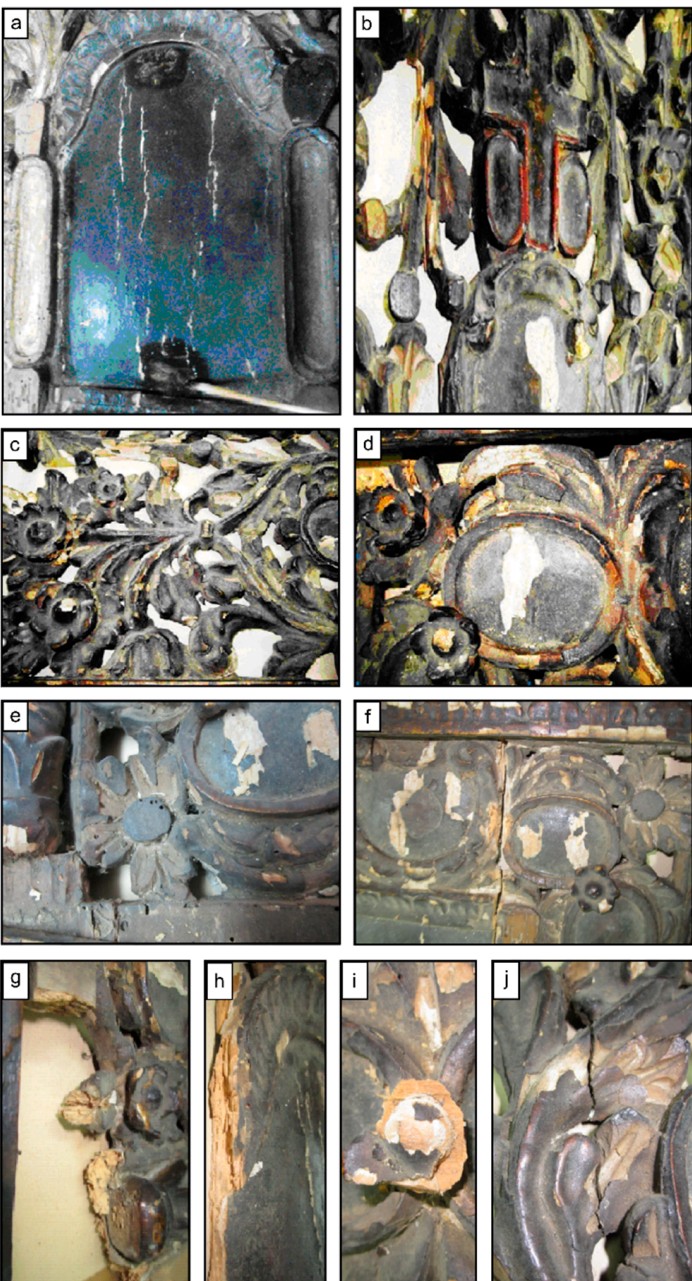

**Figure 2.** Details of the icon with deteriorations and degradations: (**a**) Degradation by oxidative fouling and deterioration by dynamic longitudinal cracks with roofing degradations in the central medallion; (**b**–**f**) Superficial and profound lacunae, oxidative incrassation and missing ornament through fraction and loss; (**d**–**f**) Central lacuna in medallions and ornament; (**g**–**j**) Advanced weakening by xylophagic attack, deep longitudinal and transverse cracks, pictorial and support materials loss.

In this regard, the scientific investigation used optical microscopy (OM) in visible light and UV, scanning electron microscopy (SEM) coupled with energy-dispersive X-ray analysis (EDX) and micro-FTIR spectrophotometry. In order to establish the state of preservation and the efficiency of the formulated/proposed washing tests, the analysis methods were corroborated with the CIE L*a*b* colorimetry, respectively, with visible reflectography, UV and IR, but also by direct observation with the naked eye.

Wood samples and pictorial material were used in the analysis, including gilded pieces, all coming from some small fragments detached from painted medallions and gilded ornaments. Furthermore, another sample was taken from the main fracture of one of the two icon fragments detached by a longitudinal crack in order to identify the wood species (Figure 1b). All samples, which were used in the analysis, also included a stratigraphic section of the painting layers (primer/bolus, polychrome layer, gilding and varnish), including dirt. These were analyzed under an optical microscope, choosing the most representative structural details of the samples. For some analysis techniques, a painting sample with a stratigraphy characteristic of the icon was used, one with dirt, two with the dominant pigments-red and blue, and two with polishes (having over the gold or silver foil and the bronze gilding restoration).

The six samples were noted as follows: P1—section in the paint layer, with preparation; P2—dirt from the gold gilded surface; P3—red pigment with preparation; P4—blue pigment with preparation; P5—silver gilding; P6—gold leaf with traces of subsequent repainting with bronze.

Following the determination of the conservation state of the main components of the pictorial materials and the composition of the dirt, we began the consolidation process.

Along with this protocol, two more were used in the restoration of the icon. The first one was used for the elaboration of the wet cleaning procedures of the polychrome surfaces and the reverse, by involving the *washing tests* and the second for the *selection of compatible materials* and the *optimal reintegration of structural and chromatic (including gilding) processes*, followed by final varnishing, with or without repatination. Compatible materials and appropriate procedures were used in these interventions, which did not affect the integrity of the old polychromes and gildings.

### 2.3. Experimental Methods and Analysis Techniques

The following techniques were used to determine the nature and conservation state of pictorial materials and gildings:

- Carl Zeiss AxioImager A1m Optical Microscope, with the possibility of working in the dark or bright field, in visible or UV, with an AXIOCAM camera attached and operating through specialized software. The samples were analyzed at magnifications between $50\times$–$500\times$. To identify pigments, fillers and dirt, microscopic analysis of reflection in polarized light was used, which provided information on the size and morphology of crystallites or pigment particles, color, homogeneity of binder dispersion, etc. The wood samples analysis was made using the same technique, by reflection, which allowed the assessment of the conservation state of the wood fiber, highlighting the anatomical elements, but also the presence of an extensive fungal or xylophagous attack [18];

- Scanning Electron Microscopy (SEM) coupled with Energy-Dispersive X-ray spectroscopy (EDX). A SEM microscope, model VEGA II LSH, produced by TESCAN Czech Republic, coupled with an EDX detector, type QUANTAX QX2, manufactured by BRUKER/ROENTEC Germany, was used in the analyses. The microscope, controlled by a computer, has a tungsten filament electron cannon that can achieve a resolution of 3 nm at 30 kV, with a magnification between $\times 30$ and $\times 1{,}000{,}000$ in the "resolution" operating mode, an acceleration voltage between 200 V and 30 kV and scanning speed between 200 ns and 10 ms per pixel. The working pressure was less than $1 \times 10^{-2}$ Pa. Quantax QX2 is an EDX detector used for the qualitative and quantitative micro analyses. The EDX detector was a third-generation, X-flash-type detector, which does not require cooling with liquid nitrogen and is approximately 10 times faster than conventional Si(Li) detectors. With the help of this technique, the nature of the pigments and the dirt were analyzed;

- TENSOR 27 micro-FTIR spectrophotometer with standard DLaTGS detector operating at room temperature. The resolution is usually 4 cm$^{-1}$, but can reach 1 cm$^{-1}$. It is equipped with a He-Ne laser that emits at 633 nm and a power of 1 mW and has a

ROCKSOLID alignment of the interferometer. The signal-to-noise ratio of this device is very good. The HYPERION 1000 microscope is a coupled accessory that works in reflection and allows completely non-destructive measurements. The system is completely controlled by OPUS software for interactive video data acquisition, which can work in both transmission and reflection. The detector is an MCT type cooled with liquid nitrogen ($-196\ °C$). The spectral range, in which the analyses were performed, is 600–4000 cm$^{-1}$ and the measured area is 1.0 mm$^2$. The microscope is equipped with a 15× lens. Both devices are from Bruker Optic, Germany. This technique determined the nature of the pigments and gildings, the type of filler of the preparations, the binders, the varnish, the nature of the dirt and the degree of its penetration;

- The evaluation of color changes due to aging and the efficiency of washing operations were performed using the LOVIBOND RT Series (Reflectance Tintometer) colorimeter. This allowed the ΔE* chromatic deviation to be recorded directly on the sample, before and after washing. The data was transferred to the computer and processed;
- Establishing the compatibility and synergy of chemical components and intervention technologies, involving separate samples of the artifact, to which accelerated aging was applied by air conditioning in ovens for thermal, hydric and predetermined conditions, followed by analysis of color and surface texture by CIE L*a*b* colorimetry, UV, visual and IR reflectography.

### 2.4. Preliminary Materials and Operations for Consolidation and Preventive Insecticide and Fungicide Application

#### 2.4.1. The Wooden Support

As the conservation state of the substrate has shown several types of deterioration and degradation, the two groups of preliminary interventions (consolidation, insecticide and fungicide) have been carried out in several stages, one preventive or temporary and the other prophylactic or definitive.

Initially, the strongly weakened part of the support was treated preventively against xylophagous attack by brushing with Permethrin 2% solution in White spirit (Figure 3).

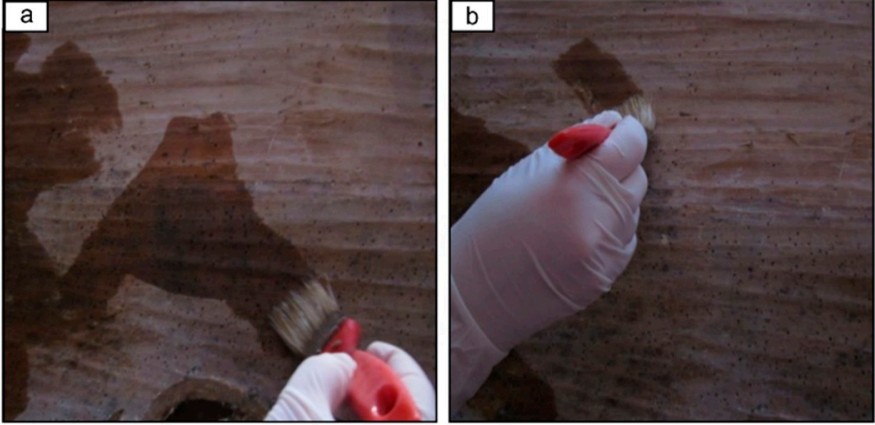

**Figure 3.** Preventive treatment against xylophagous attack by brushing with 2% Permethrin solution (**a**,**b**).

After the application of the insecticide and fungicide, the strongly weakened part of the substrate was impregnated by injection and brushed with solutions of Paraloid B 72 in Xylene of different concentrations (10%, 15% and 25%), depending on the degree of damage and porosity of the areas (Figure 4).

Seven days after impregnation, the two detached parts following the longitudinal fracture of the carved panel were glued with a warm aqueous dispersion of 40% bone glue, by tightening, using a system of three metal vices (Figure 5a) and four plastic clamps used to fix the fractures (Figure 5b). This operation was performed after linden plugs (Figure 6a) were previously implanted on the solder area in the corresponding holes drilled with an

auger with a diameter equal to the plugs (Figure 6b), and then in the contact interfaces were applied thin strips of balsa wood to the plugs (Figure 6c).

The filling of the lost wood mass from the lower right corner due to the weakening by intense xylophagous attack was conducted with sawdust grout and 15% bone glue, on a reinforcement made of bamboo sticks and flax oakum (Figure 7).

To remove the dirt, grease and soot deposits from the reverse of the icon, cleaning was performed with a 10% technical alcohol warm solution in distilled water. After cleaning the reverse, in order to prevent the degradation of the wood by the attack of insects and to protect it from humidity, a layer of natural wax mixed with colophony in a ratio of 3:2 was applied. The final layer was polished by rubbing with a cotton cloth. This operation on the cleaned reverse has been applied after the structural integration operations by completing, gluing and consolidating the ornate support and the painting layer. The surface of the reverse was impregnated with a mixture of beeswax and colophony melted with a hot air blower (Figure 8a), and the carved or perforated areas of the support with the electric spatula (Figure 8b).

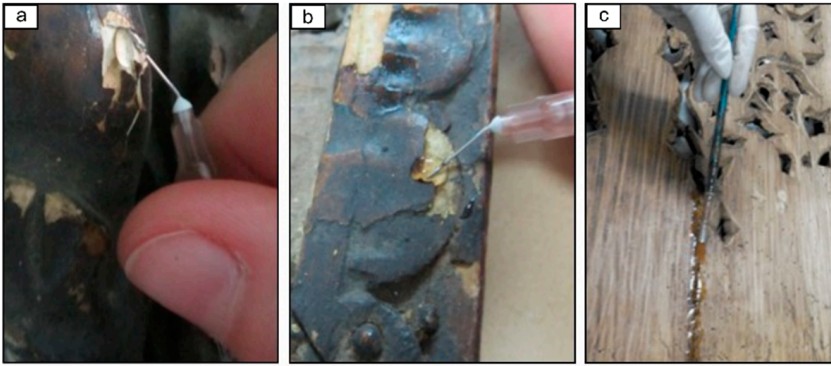

**Figure 4.** Consolidation in the volume phase of the weakened support: (**a**,**b**) Injection; (**c**) Brushing the Paraloid solution B 72.

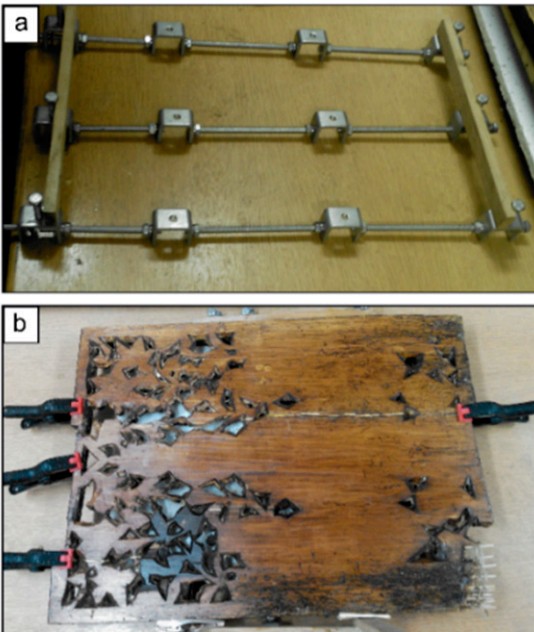

**Figure 5.** Fastening and clamping systems with metal and plastic vices: (**a**) The tightening system through three metal vices; (**b**) Fixing with plastic clamps.

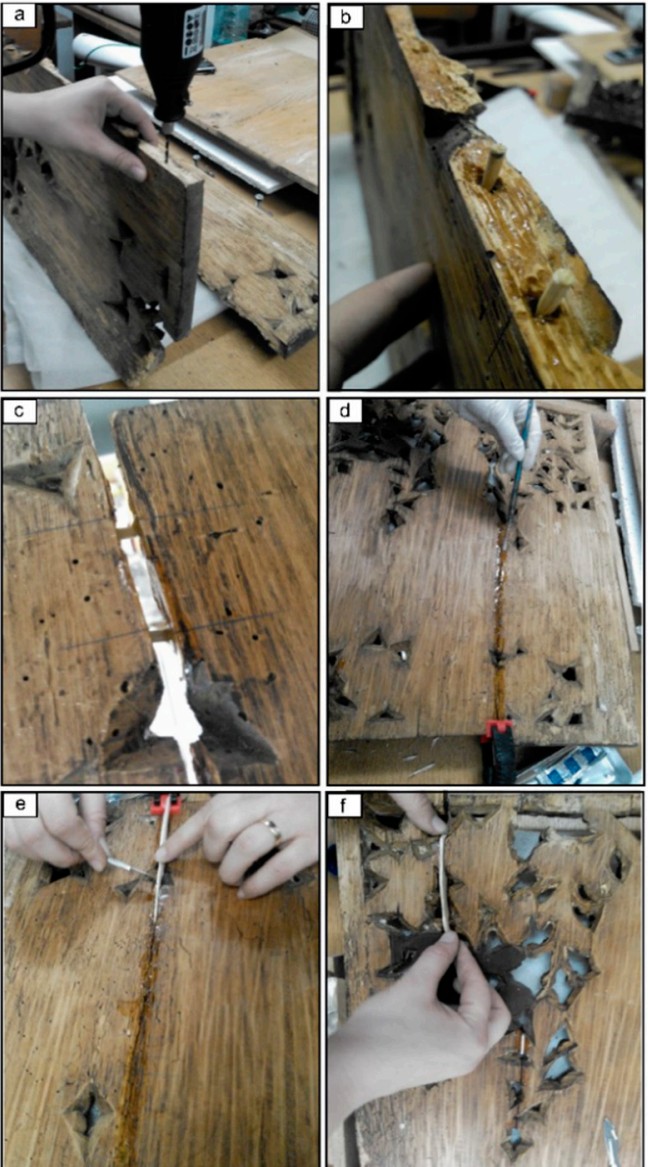

**Figure 6.** Aspects regarding the stages of reintegration and structural stabilization interventions: (**a**) Drilling holes for plugs; (**b**) Implanting the plugs in the soldered area of the two detached parts of the panel; (**c**,**d**) Fixing and gluing the two fractured fragments; (**e**,**f**) Application of balsa wood strips in the cracks areas before tightening into the vice.

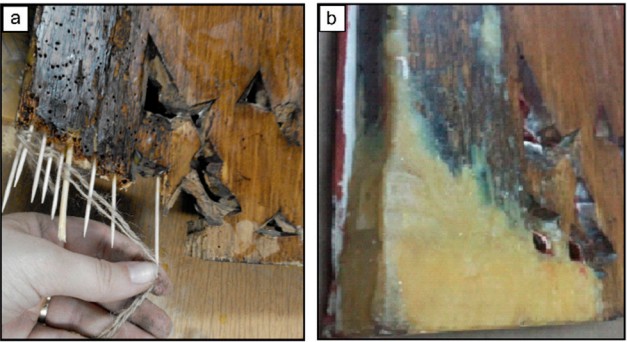

**Figure 7.** Installation of the reinforcement made from bamboo sticks and flax oakum (**a**) in the filling area by sawdust grout (**b**).

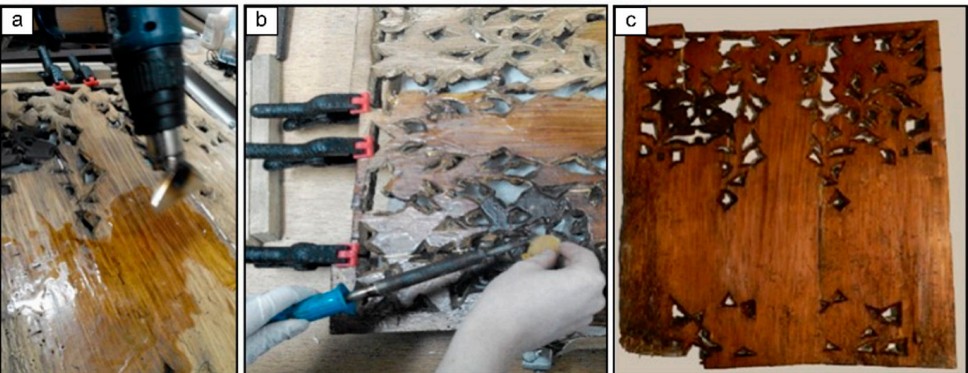

**Figure 8.** Hot impregnation with beeswax and colophony: (**a**) Impregnation of the surface of the reverse with a hot air blower; (**b**) Impregnation of carved and perforated areas with an electric spatula (**c**). The reverse image after the total impregnation.

The lost ornamental elements were made of balsa wood, carved in the shape of the initial profile (Figure 9), which, after grouting and making the bolus preparation, were glued separately on the missing areas, using warm aqueous dispersions of 5% fish glue.

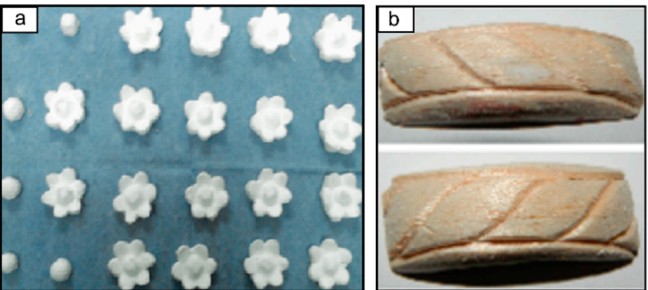

**Figure 9.** Lost ornamental elements made of balsa wood: (**a**) Buttons and ornamental flowers; (**b**) Capitals and plinths of the columns.

### 2.4.2. Pictorial Layer

In the case of the pictorial layer, the preliminary interventions of consolidation and preventive fungicization were carried out in several stages. Thus, for the temporary/preventive consolidation of the painting layer, Japanese hot-impregnated paper was used with a dispersion of fish glue of over 4%, in distilled water, containing 2% technical alcohol and 2% phenol, over which Melinex thermal insulation foil was applied and pressed with a warm spatula (38 °C). After wet cleaning, the final consolidation was conducted using 5% aqueous dispersion of fish glue by hot brushing on the entire surface.

The lacunar areas were chromatically reintegrated by the *tratteggio* technique using water colors and egg emulsion for both painting and gilded ornaments. These were applied differently only on the painted areas and on the existing gilded ornaments (unaffected by the bronze applied later). Initially, the gaps were degreased with a 10% technical alcohol solution in distilled water and evenly distributed with a brush, using a warm aqueous dispersion of 5% fish glue in order to achieve a good adhesion of the grout.

The multilayer grouting was performed with a concentrated dispersion of chalk powder and 5% fish glue, in a mixture of 2% technical alcohol, 2% phenol, 1% bee honey and 2% Damar resin solution in 15% turpentine, the last components having the role of increasing processability and preventing fungal attack. The dry grout was finished with a scalpel and abrasive paper (sandpaper), in the presence of a thin layer of 2% acetic acid, then the pores were leveled with shellac. After the shellac dried, the chromatic reintegration was performed. In order to imitate gold and its reflections, on the areas that did not allow regilding, interposed lines were used with ocher, yellow, green and red in equal proportions (*trattegigo* technique).

The missing ornamental elements and the rod on the right edge (initially fixings for the icon in the box) were carved of balsa wood, using as a model those that have been preserved. After making the desired profile for each element (seven buttons, nineteen flowers, two capitals, two consoles and one rod), a layer of 5% fish glue was applied, then a preparation of several layers of chalk powder with 5% fish glue, 2% technical alcohol, 2% phenol, 1% bee honey and 2% turpentine solution with 15% Damar resin. After sanding the preparation with a scalpel and sandpaper (sandpaper), in the presence of a thin layer of 2% acetic acid, the pores were leveled with shellac, over which, after drying, was applied a layer of bolus (red-brick colored clay). After polishing with an agate stone, the gold foil with a Mixion adhesive was applied, which dries in approximately 15 min. After gilding, these elements were brushed with a thin layer of burnt umber in boiled flax oil (varnished), and finally varnished along with the entire icon by spraying with shellac mixed with a little burnt umber.

After the wet cleaning of the old gold and silver gildings, using a solution based on 50% turpentine, 20% absolute ethyl alcohol, 15% ethylene glycol and 15% cyclohexanol, the previous bronze regildings were carefully removed (performed by an unauthorized intervention), using an aqueous dispersion of Contrad 10% (commercial product based on alkaline aqueous emulsion of surfactants). Then, the grouting and chromatic reintegration were made.

*2.5. Cleaning Systems and Washing Test*

Elaboration of the Cleaning Recipe through the Washing Test

The icon of the "Coronation of the Virgin Mary" presented, as we said, a uniform layer of heavy dirt, adhering to both the painting substrate and the gilding, having a glassy, black-brown appearance. Moreover, some ornaments were regilded with bronze through an empirical restoration about 100 years ago.

The specialized literature and the practice of the authors [26–29] recommended the usage of an organic solution for oil paintings varnished with a high degree of dirt, already optimized for other previous interventions. For this solution, based on 50% turpentine, 20% absolute ethyl alcohol, 15% ethylene glycol and 15% cyclohexanol, it was no longer necessary to apply the wash test.

Instead, for the gilded ornaments which showed older bronze gilding, in addition to the dirt, it was necessary to optimize dispersion for wet cleaning. Thus, one of the frame rods with extended planimetry, which had uniform deposits of dirt and bronze, allowed the determination of chromatic deviations following the cleaning operations when applying the washing tests.

For this purpose, eight solutions of ethyl alcohol mixed with other organic solvents and two commercial products designated for such deposits were used. From these, the most effective compositions were selected, which did not affect the gold gilding, the old patina and the unaltered varnish. The washing was performed by cross-wiping with cotton swabs soaked in the washing solution, in several stages until optimal cleaning.

In the process of cleaning the embedded dirt, in order of frequency of use for oil paintings and for old varnished gildings, the following groups of systems were used for the washing test, marked with letter E:

E1—ethyl alcohol + dichloroethane (8:1);
E2—ethyl alcohol + turpentine (1:1);
E3—ethyl alcohol + turpentine + cyclohexane (1:1:1);
E4—ethyl alcohol + turpentine (8:1);
E5—ethyl alcohol + cyclohexane (8:1);
E6—ethyl alcohol + water (7:1);
E7—commercial product type Kromofag (based on dichloromethane and methyl alcohol 2:1);
E8—commercial product type Contrad (alkaline aqueous emulsion of surfactants);
E9—ethyl alcohol + xylene (2:1);

E10—ethyl alcohol + xylene (4:1).

Of these, some are recognized as having some aggression on old degraded paintings, such as the commercial product type Kromofag (CT7) and Contrad (CT9), the latter being extremely toxic to the restorer. These two products were accepted in the protocol as they allow a wide range of comparison, being on the list of classic washing systems of strongly adherent organic deposits.

## 3. Results and Discussion

### 3.1. The Nature of the Materials and Their Preservation State

By means of the optical microscopy and the keys for the microstructural identification of the anatomical parts of the wood, it was established that the icon panel is made of linden (*Tilia cordata Mill*). Involving stereomicroscopy and UV, visual and IR reflectography, the conservation state of the wooden support was analyzed, along with the pictorial layer and the gildings. It has been noticed that the icon is in a reversible precollapsed state, with advanced deterioration and degradation in some areas, close to almost irreversible collapse.

Furthermore, the optical magnifying glass was used to analyze the sampling areas of the pictorial material and gildings. The six samples of pictorial material were initially analyzed under an optical microscope.

As can be seen from Figure 10a,b, the dirt on the surface of the painting is opaque with a rough microstructure, in which dust and soot particles are trapped. Furthermore, small drops of oil and wood powder from woodworms are noticeable. The third sample (Figure 10c), taken from the edge of an area with a superficial gap in the pictorial layer, is represented by an intense red pigment with small bronze traces. The fourth image (Figure 10d) is a fragment from a medallion background, light blue, covered with flax oil and dark varnish.

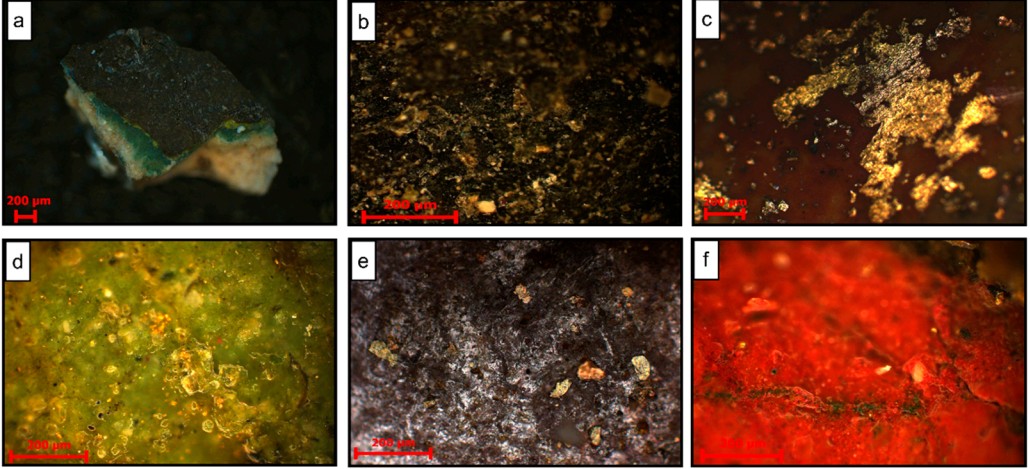

**Figure 10.** Microphotographs of the pictorial layer (50–200×): (**a**) Section in the paint layer, with the preparation, colors, varnish and embedded dirt; (**b**) Varnish with embedded dirt and bronze gilding traces; (**c**) Red pigment, varnish and bronze gilding traces; (**d**) Blue pigment with slightly stained varnish; (**e**) Silver foil with bronze gilding traces; (**f**) Gold leaf with bronze gilding traces.

The sample with the silver foil (Figure 10e), taken from the center of a flower, had superficial particles from the last bronze regilding, traces of dust and dirt, and the sample with the gold foil (Figure 10f), collected from a flower petal, has a worn surface as it comes from a curved area of the sculpture, where there are traces of bronze and embedded dirt.

Impurities are present in all samples, due to dirt and bronze gilding, that have penetrated into the varnish, and the painted surface has irregularities in all samples.

The SEM microphotographs (Figure 11) of the six samples of the pictorial material and gildings, allowed the highlighting of the surface distribution and morphology of their structural components.

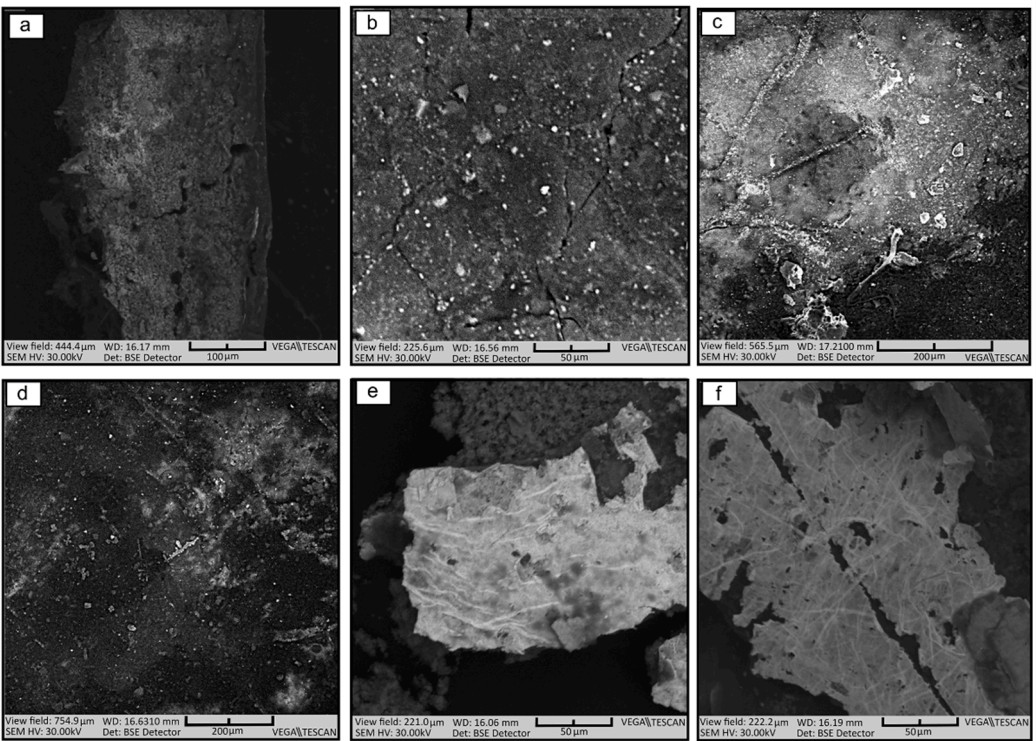

**Figure 11.** SEM microphotographs in BSE (200–1000×): (**a**) Preparation; (**b**) Varnish and embedded dirt; (**c**) Red pigment; (**d**) Blue pigment; (**e**) Silver foil with bronze marks; (**f**) Gold foil with bronze marks.

The preparation was analyzed on a clear area in the stratigraphic section of the pictorial material from P1 sample (Figure 11a), which shows discontinuities and some fine longitudinal cracks in the center. The analysis of the varnish from the gilded surface (Figure 11b) highlights the traces of embedded dirt, strongly penetrated into the varnish, that has fine cracks on the surface, traces of bronze gilding, soot and atmospheric dust, with uneven distributions and granulations. The red pigment in P3 sample (Figure 11c) and the blue pigment in P4 sample (Figure 11d) also show soot and atmospheric dust traces, with uneven distributions and granulations. The silver foils in P5 sample (Figure 11e) and the gold ones in P6 sample (Figure 11f) presents discontinuities, gaps, and erosion iridescence with uneven varnish, traces of bronze gilding and surface structures embedded on certain areas with soot and atmospheric dust. The P2 sample (Figure 11b), which also has traces of bronze gilding, was analyzed by EDX separately from the areas with these marks.

Thus, the EDX analysis established the elemental composition of both the preparation, the varnish layer with dirt, the two basic pigments (red and blue), as well as the two types of gilding with silver and gold foils (Table 1).

In the preparation (from P1 sample) were identified the components of the primer: Ca, C and O based on chalk powder ($CaCO_3$) and a series of microelements (Si, Al, Mg, Na, K and Fe) specific to the red-brown bolus. The fact that the mass ratio of the chemical elements corresponds to the composition of the chalk powder and the red earth for the bolus, indicates that the preparation was well preserved.

Regarding the degraded varnish (from P2 sample), it is inserted with soot dirt (C, O and S) and atmospheric dust (Ca, Mg, Na, K, Fe and Cl).

The red pigment (P3 sample) is based on cinnabar, which contains, in addition to mercury and sulfur, a number of elements specific to colored earths (Si, Al, Ca, Mg, Na, K and Fe), that were mixed with lead Minium (Pb and O) and ceruse (Pb, C and O) to give brightness to the color [18,19,33].

The blue pigment (P4 sample) is based on ultramarine or Lapis Lazuli (Si, Al, Mg, Na, K, Fe and Cl), applied on preparation layers based on chalk powder and gypsum (C, O, S and Ca).

**Table 1.** EDX elemental composition of the analyzed samples of pictorial materials and gildings.

| Chemical Element | Elemental Composition in Weight Percent (%): | | | | | | |
|---|---|---|---|---|---|---|---|
| | Preparation from P1 Sample | Varnish + Dirt from P2 Sample | Red Pigment from P3 Sample | Blue Pigment from P4 Sample | Ag Foil from P5 Sample | Au Foils from P6 Sample | Bronze Gilding (Regilding from P2 Sample) |
| Au | - | - | - | - | 10.91 | 46.33 | - |
| Ag | - | - | - | - | 47.27 | 3.54 | - |
| Cu | - | 1.95 | - | - | - | 1.92 | 59.31 |
| Zn | - | 0.53 | - | - | - | 0.61 | 17.66 |
| Hg | - | - | 20.02 | - | - | - | - |
| Pb | - | - | 26.43 | - | - | - | - |
| C | 11.36 | 20.08 | 10.43 | 18.74 | - | 11.95 | 9.27 |
| O | 45.38 | 64.54 | 22.57 | 64.46 | 30.31 | 24.73 | 13.75 |
| S | - | 0.70 | 8.21 | 4.47 | - | - | - |
| Si | 1.36 | 2.37 | 2.23 | 2.98 | - | 2.72 | - |
| Al | 0.72 | 0.86 | 3.21 | 1.40 | - | 2.39 | - |
| Ca | 38.03 | 3.73 | 2.57 | 2.54 | 9.22 | 3.62 | - |
| Mg | 0.54 | 0.36 | 1.00 | 0.94 | - | - | - |
| Na | 1.15 | 1.56 | 0.89 | 2.49 | - | - | - |
| K | 0.44 | 1.98 | 0.53 | 1.01 | - | - | - |
| Fe | 1.02 | 1.01 | 1.86 | - | - | 2.19 | - |
| Cl | - | 0.33 | - | 0.91 | 2.30 | - | - |

The silver foil (P5 sample) is made of an Ag alloy with about 11% Au, and the gold one (P6 sample) has a low content of Ag, about 3.5%, along with traces of regilding with fine brass powder (Cu and Zn), which contains traces of degraded and soot varnish (C, O, Si, Al, Ca and Fe).

Regilding traces, analyzed on a representative (well-preserved) area of P2 sample, are represented by thin layers of bronze that were applied after the manufacturing of the icon by empirical restoration interventions. These layers are based on fine brass powder (Cu and Zn) and contain traces of degraded varnish and soot.

The presence of cinnabar in the composition of the red pigment, with traces of Minium and red earth, shows that the painter used the natural Vermillion, which also indicates the antiquity of the icon, most likely made in the first half of the 19th century [16,17,33,52]. Some of the trace elements come from the preparation (chalk powder), varnish and dirt (soot and atmospheric dust).

In addition to Vermillion and ceruse, the red pigment also contains burnt Sienna (colored earth used to make shadows and bolus). This is marked by the presence of iron, aluminum, silicon, magnesium, sodium and potassium [16,17,33].

The chemical composition of the blue pigment is close to that of Ultramarine (Lapis Lazuli) with the molecular formula $Na_8Al_6Si_6O_{24}S_4$, with some impurities from the dirt and primer [16,17,33].

Based on the above premises regarding the nature of the pictorial materials, evaluated from the elemental composition EDX (Table 1), some of the samples were correlated with the μ-FTIR analysis. Thus, out of the six pictorial materials analyzed by SEM-EDX, only three, that absorb IR, were investigated by μ-FTIR; the preparation (from the area with missing gilding, which contains bolus and chalk powder-P2 sample) and the red (P3 sample) and blue (P4 sample) pigments.

The values of specific group vibrations in the two regions of the FTIR spectrum, 4000–1500 cm$^{-1}$ specific for valence vibrations (υ) and 1500–600 cm$^{-1}$ for deformation vibrations (δ) [19,27–29,40,41] are shown in Table 2.

By corroborating the results obtained by SEM-EDX and μ-FTIR analyses, their conservation state was established, besides the precise confirmation of the nature of the three pictorial materials and their evolution in time, after the manufacturing of the icon.

**Table 2.** Specific group vibration values of the bands for the three samples of pictorial material (preparation-P2, red pigment-P3 and blue pigment-P4).

| Samples | Functional Groups | Specific Group Frequencies (cm$^{-1}$) | |
|---|---|---|---|
| | | Valence Vibration Region (υ) 4000–1500 cm$^{-1}$ | Deformation Vibration Region (δ) 1500–580 cm$^{-1}$ |
| P2 | Dried and aged boiled flax oil | 3470/i, 2964/i, 2874/i, 1942/w, 1747/i | 1469/i, 1322/i, 796/m, 765/I, 723/i, 631/i |
| | Animal glue | 2210/w, 2081/w, 1986/w, 1678/m, 1573/i | 1056/i, 916/i, 834/m, 723/i, 923/m |
| | H$_2$O, -OH, -O$^-$H$^+$ | 3978-3945/w, 3764/w, 3620/m, 1678/w | - |
| | Chalk powder | 2514/w | 1086/m, 907/m, 713/w |
| | Bolus (ceruse, minium, colored earth) | 3696/m | 1352/m, 1054/m, 853/i, 622/w |
| P3 | Dried and aged boiled flax oil | 3424/i, 2958/i, 2865/w, 1743/i, | 1477/i, 885/i, 856/i, 835/i, 797/m, 727/m |
| | Animal glue | 2518/m, 2093/i, 1515/m | 1393/i, 1305/i, 884/i, 690/i |
| | H$_2$O, -OH, -O$^-$H$^+$ | 3944-3915/w | - |
| | Chalk powder | 2523/w | 1450/m, 1086/m, 870/m, 799/m, 712/i |
| | Cinnabar or Vermillon (HgS) | 3318/m, 3300/m, 2205/w | 642/m |
| | Minium (Pb$_3$O$_4$) | 1526/m | 1223/m, 925/m |
| P4 | Dried and aged boiled flax oil | 3547/i, 2975/i, 2877/i, 2675/i, 2602/i 1747/i | 1468/i, 887/m, 797/m, 697/m, |
| | Animal glue | 2093/i, 1998/m, 1958/m, | 1394/i, 1286/i, 1196/i, 1138/i |
| | H$_2$O, -OH, -O$^-$H$^+$ | 3954-3922/w | - |
| | Chalk powder | 2512/w | 1425/m, 1409/m, 875/i, 713/w |
| | Lapis Lazuli (Na$_8$Al$_6$Si$_6$O$_{24}$S$_4$) | 2921/i, 2650/m, 2232/m, 1620/m, | 1352/m, 1098/m, 1054/m, 1000/m, 907/m, 853/i, 655/i, 622/w |

Note: **i.** intense; **m.** medium; **w.** weak.

The spectra of the three samples have a similar complexity, as pigments and fillers are closely dispersed in the two groups of binders: dried boiled flax oil, very poorly saponified, for red and blue colors and the animal glue for the chalk powder preparation and red bolus from earth colors.

The characteristic peaks of the old flax oil in P3 sample (red pigment) corresponding to the two ranges 4000–1500 and 1500–600 cm$^{-1}$ are: 3424, 2958, 2865 and 1743 cm$^{-1}$, respectively, 1477, 885, 856, 835, 797, 727 cm$^{-1}$, which, as a result of aging and poor saponification processes due to the lead-based colors (Minium and ceruse) differ in value for the P4 sample, with peaks: 3547, 2975, 2877, 2675, 2602 and 1747 cm$^{-1}$, respectively, 1468, 887, 797,697 cm$^{-1}$. At 1743 cm$^{-1}$, for P3 sample, and 1747 cm$^{-1}$, for P4 sample, are the representative peaks of the ester, which is attributed to fatty acids formed by hydrolysis of glycerin esters, whose degradation is limited by oxygen access [16,33,53]. The same peaks with small value changes of the wave number and intensity are found in P2 sample: 3470, 2964, 2874, 1942, respectively, 1469, 1322, 796, 723 and 631 cm$^{-1}$. The representative peak of the ester, attributed to the fatty acids formed by the hydrolysis of chemically modified glycerin esters by aging, is found in P2 sample at 1747 cm$^{-1}$. Similarly, the peaks specific to the animal glue from the preparation, in the form of traces, are found in all three samples, with some changes in the wave number and the intensity values.

The spectrum appearance for Vermilion or Cinnabar overlaps with that of the boiled flax oil from the P3 sample composition, with very small differences in the specific area of the deformation vibrations (δ) at 642 cm$^{-1}$. The Vermilion also has representative peaks that overlap with those of the waters (3318, 3330 and 2205 cm$^{-1}$), then over those of the ceruse and Minium at 1526, 1223 and 925 cm$^{-1}$. The relatively intense peak at 1526 cm$^{-1}$ is attributed to lead soap [16,33].

The IR spectra for ultramarine pigment (Na$_8$Al$_6$Si$_6$O$_{24}$S$_4$) show specific absorption bands: aluminosilicates (1352, 1098, 1054, 1000, 907, 853, 655 and 622 cm$^{-1}$), calcite (1450 and 870 cm$^{-1}$) and residual kaolinite (907 cm$^{-1}$). The much narrower and strongly diminished band from 3954–3922 cm$^{-1}$/w, together with the very weak peaks from 3696 and 3620 cm$^{-1}$, which belong to the different forms of water (hygroscopic, crystallization and constitution), present as neutral waters, weakly basic or slightly acidic (H$_2$O, -OH, -O-H$^+$) eliminated by prolonged drying over time, have a very low spectrum, being barely perceptible/noticeable.

### 3.2. Efficiency of the Washing Tests and the Icon Cleaning

An organic solution was used for painting, previously optimized according to former studies [25–29], using the organic mixture based on turpentine, absolute ethyl alcohol, ethylene glycol and cyclohexanol (1:1:1:1), and did not require washing tests. Instead, a wet cleaning system was applied to the old dirty gilding, with bronze based on fine brass powder, which was intended to optimize the wash test. The work area chosen was represented by the upper rod of the frame, which presented a layer of embedded dirt with uniform distribution and structure/morphology. The ten wet cleaning solutions were applied here. Moreover, it is known that the gold gilded ornaments are resistant to a large number of cleaning stages by wiping, which allowed the extension of this process until optimal cleaning, without damaging the old patina and undamaged varnish. Figure 12 shows the areas selected for the application of the ten wash tests.

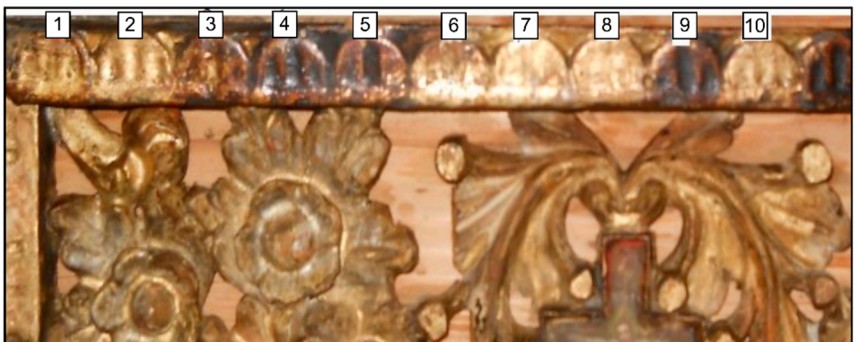

**Figure 12.** Washing tests for the gold gilded wood carving and embroidery with mono or mixed organic solvent solutions (E1–E10).

Using direct observation (Figure 12) and CIE L*a*b* colorimetric analysis (Figure 13), the wet cleaning efficiency was established for all the studied systems, based on the wash test, in the following descending order: **E2 > E7 > E8** > E10 > E6 > E1 > E3 > E5 > E4 > E9.

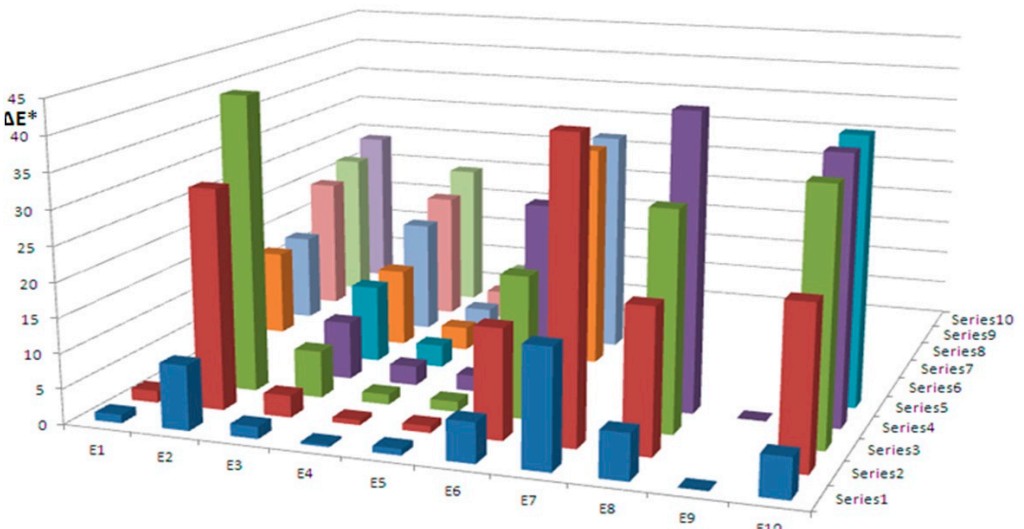

**Figure 13.** Cleaning efficiency by CIE L*a*b* colorimetry with E1–E10 systems (ΔE* is the chromatic deviation).

If we consider the previous observations and the data in Table 3, the E2 system was used to clean the gildings, as after only three washing steps it effectively cleaned the dirt deposit, without affecting the old patina and the undegraded varnish.

**Table 3.** Efficiency of wet cleaning systems in the wash test.

| Washing Test | Components | Volumetric Ratio | Number of Washing Steps | Period Required for Final Cleaning (min) | Observations (Washing Efficiency) |
|---|---|---|---|---|---|
| E1 | Ethyl alcohol + dichloroethane | 8:1 | 12 | 5 | weak |
| E2 | Ethyl alcohol + turpentine | 1:1 | 3 | 3 | rapid |
| E3 | Ethyl alcohol + cyclohexane + turpentine | 1:1:1 | 9 | 3 | weak |
| E4 | Ethyl alcohol + turpentine | 8:1 | 8 | 5 | verry weak |
| E5 | Ethyl alcohol + cyclohexane | 8:1 | 9 | 5 | verry weak |
| E6 | Ethyl alcohol + water | 7:1 | 7 | 5 | rapid |
| E7 | Kromofag commercial product | - | 2 | 2 | rapid |
| E8 | Contrad commercial product | - | 4 | 2 | rapid |
| E9 | Methyl alcohol + xylene | 2:1 | 2 | 2 | weak |
| E10 | Ethyl alcohol + xylene | 2:1 | 5 | 3 | medium |

### 3.3. Structural and Chromatic Reintegration, Regilding and Revarnishing with Repatination

The multiple cracks and total or partial detachments of the pictorial layer and the missing parts of the gilded floral elements required preventive consolidation interventions, followed by the cleaning operations, applied differently on the pictorial and gilding layers. The preventive consolidation was conducted according to a traditional method by hot layering of a 5% aqueous solution of fish glue, in distilled water, over the Japanese paper, placed on the areas with detachments of the pictorial layer. For the polychrome areas, for leveling the detachments, it was necessary to apply a concentration of 8% fish glue in distilled water. The layering was applied from the center to the edges, radiating, to keep the Japanese paper as smooth as possible. After the brushing and interposing a Melinex heat-resistant foil, the adhesion of the paint layers and the detached gildings was achieved by applying warm sandbags in the consolidated areas. After that, the heat-resistant foil was removed and the excess glue was eliminated with absorbent paper.

Subsequently, tests were carried out to remove the dirt layer (washes) from the carved polychrome with: absolute ethyl alcohol; xylene; cyclohexanol; turpentine; dichloroethane; Contrad; and Kromofag (the last two are non-ecological commercial products), in the form of organic systems (Table 3). The dirt was easily removed with the application of aqueous emulsion of 90% ethyl alcohol (E6), Contrad (E8) and Kromofag (E7), these three systems being taken as a reference, as they also allowed the removal of fatty, oily and proteinaceous substances from the gilded areas. In order to dissolve the varnish layer and to facilitate the action of the Contrad and Kromofag dispersions, it was first layered with absolute ethyl alcohol using a brush. These substances were applied by wiping with small cotton swabs, dirt being removed successively by circular or cross moves. It should be noted that the dirt on the painted medallions disappeared easily by the action of a mixture of 70% turpentine and absolute 30% ethyl alcohol (E2), applied by the same method as in the case of polychrome areas, being one of the effective washes, and which, along with E6, has the advantage of very low toxicity compared to Contrad and Kromofag (non-ecological) commercial products.

After the removal of the dirt layer from both the polychrome and gilded areas, the lacunar areas, which covered most of the icon, were filled. The superficial or deep gaps were loaded with a grout consisting of $CaCO_3$ and fish glue 5%. The grout was prepared using a dispersion of 5% fish glue in distilled water, heated on a bain-marie, over which was added the chalk powder until the mixture was of a thick consistency. Subsequently, were added a few drops of phenol, biocide for microorganisms and xylophagous insects and a little honey with a plasticizing role. The application of the grout was made with a brush, after, and beforehand the hot glue was brushed over the lacunar areas, in order to facilitate the grout adhesion to the support.

The missing frame ornaments and phytomorphic elements, made of balsa wood, were grouted and chromatically integrated and then varnished. During the interventions the icon was kept face up in a room with an optimal microclimate.

In some large areas, paint and gilding were having multiple cracks and detachments. These were first fixed preventively with 5% fish glue and Japanese paper (Figure 14), after which the actual (final) consolidation interventions were performed. As the adhesion of the paint layers to the substrate on certain areas was not achieved from the first application, successive consolidations were necessary, as well as the increase in fish glue concentration (8%).

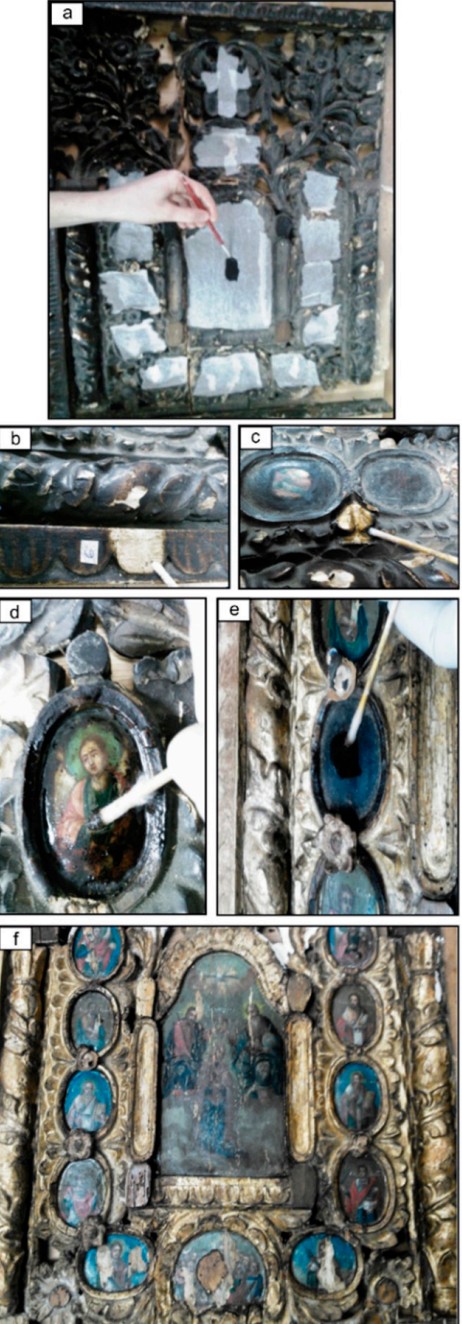

**Figure 14.** Preventive consolidation and cleaning of various forms of dirt: (**a**) The consolidation of the painting layer; (**b**,**c**) Removal of the dirt with ethanol and Contrad from the polychrome surface; (**d**,**e**) Details from the removal of adherent and semi-adherent dirt from the painting layer; (**f**) Removal of adherent dirt from polychrome surfaces and extended gilding (final detail).

Following the cleaning of the dirt deposits (Figure 14a–c), the shine of the gold gilding alternated with the silver one highlighting the richness of phytomorphic elements that harmoniously connect the medallions of the Saints with the central scene of the "Coronation of Virgin Mary".

The removal of dirt from the pictorial layer (Figure 14d–f) revealed that the Saints clothing has vivid colors, whose beauty was hidden from viewers by the thick layer of adherent dirt deposits.

In order to restore the physical-structural integrity of the icon and its historical message, the superficial and deep gaps were cleaned and, after the layering with 5% fish glue for better adhesion of the new primer with the support, primer was applied with successive brushing of the gaps. To prevent cracks in the new primer, the new layers were applied only after each one had dried well (Figure 15). The chromatic integration was completed in water colors, in the *tratteggio* technique (Figure 16).

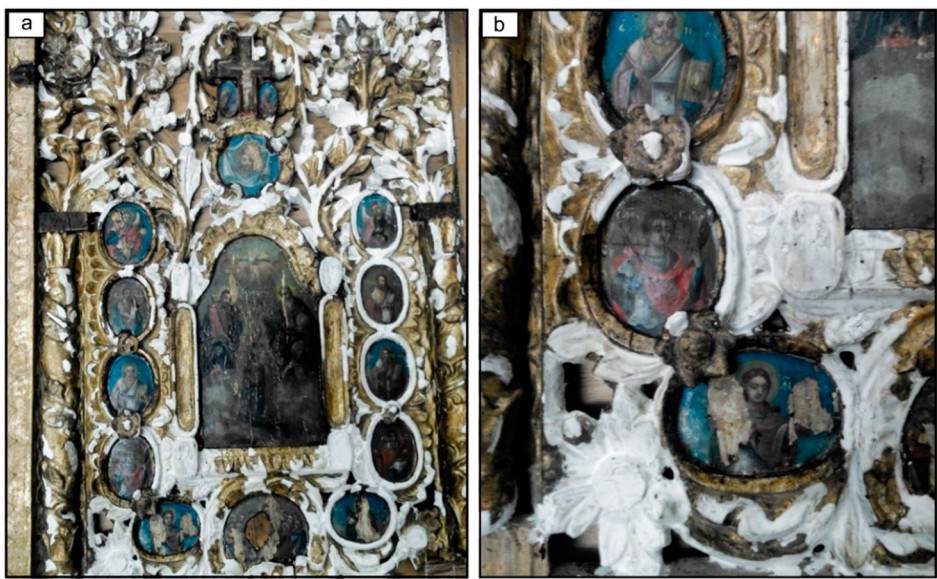

**Figure 15.** The final filling: (**a**) overall appearance; (**b**) detail.

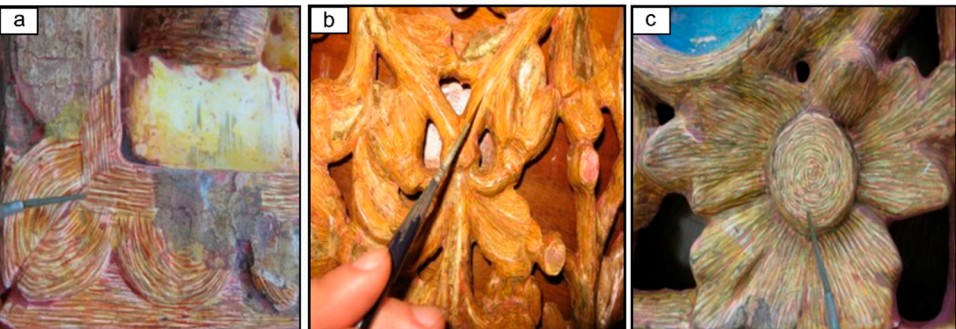

**Figure 16.** Details of the final gilding reintegration in the *tratteggio* technique: (**a**–**c**) details.

Figure 17 shows a comparison of the two stages: the final cleaning and grouting of lacunar areas with materials compatible with the preparation (Figure 17a), along with revarnishing and partial repatination of the painted areas over the reintegration by gilding in the *tratteggio* technique.

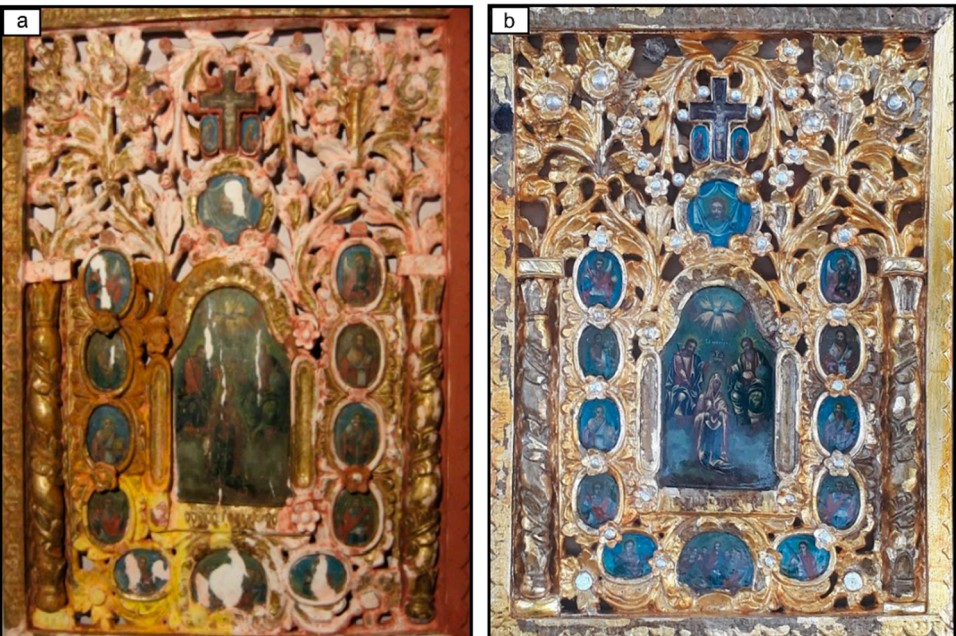

**Figure 17.** Final gilding reintegration in *tratteggio* technique: (**a**) Final cleaning and grouting of the lacunar areas with materials compatible with the preparation; (**b**) Revarnishing with partial repatination over the reintegration by gilding and the *tratteggio* technique of the painted areas.

Other operations carried out were the restoration of the lost carved ornamental elements, their priming and fixing. For this purpose, balsa wood was used, in overlapping strips and glued together with 20% bone glue, after which they were carved imitating the original shapes. Before priming, the elements were graved and brushed with 6% fish glue, for a better adhesion of the primer. The priming was carried out by gradual brushing, with drying times between layers, with an important role in the restoration of the connection between the wooden support and the painting layer. Sanding was conducted with abrasive paper. To close the pores, but also to insulate the grouted surface, ornamental elements were coated with shellac.

The chromatic integration of the primed areas and of those with use-wear traces of the pictorial layer was achieved in an imitative way, using oil colors mixed with varnishes. The chromatic integration represents an aesthetic approach that proposes the reintegration of the fragmented pictorial field, giving the viewer the opportunity to see the complete image of the icon. The choice of the aesthetic presentation method was justified by the reversibility, and by the compliance of the minimum intervention principle with the preservation of the old patina.

Before performing this operation, we apply a few layers of shellac dissolved in technical alcohol (14%—140 g of shellac flakes in 860 mL of technical alcohol). The role of this layering is to prevent excessive absorption of the Mixtion by the primer. The Mixtion is a water-based special adhesive for gold and silver foil, with a drying time of 15 min. After this process, the gold foil is applied using a brush, which must have long and soft hair, so that the foil could be stretched, removing the excess and eliminating the air bubbles. The gold foil used is of superior quality, each one being individually packed up and much thicker than the transfer one. During the application process it was placed on a paper sheet and then glued to the desired area. Furthermore, during the removal operation of the adherent and semi-adherent dirt, it was noticed that some elements were silvered; a process that was performed in order to reproduce the chromatic "game" gold—silver.

After the gilding operation, the entire surface was coated with damar varnish dissolved in turpentine (10% damar resin solution). At this point, the gold foil had a new luster effect, giving the whole assembly a non-unitary, patchy look. For this reason, we used the method of patinating the new sheet with varnishes in shades of ocher–lemon yellow–brown (1:1:1),

trying to imitate the original aspect. These colors (Idea VETRO) are solvent-based and are stable in the presence of light, durable and adaptable to temperature variations. The used varnishes are transparent so as not to damage or stain the gold foil, allowing the reflection of gold from under the layer. The colored layers are transparent enough to render as clear–obscure. The beauty of this technique is that the elements seem to have their own luster, which radiates through the layered surface. After lacquers dried, the entire surface was varnished with 10% damar turpentine.

## 4. Conclusions

Due to the precarious preservation state of the icon, close to precollapse, it was necessary to stop the effects of deterioration and degradation, a very difficult task that requires multiple interventions, both in the prophylactic preservation stage (cleaning, insecticide, etc.), as well as in the restoration one (grouting, completing structural elements, chromatic reintegration, regilding, revarnishing), in order to recover the structural–functional and aesthetic–artistic integrity, reestablishing the initial message of the icon.

All these aspects required the elaboration of an innovative experimental protocol, in an interdisciplinary system, which includes a series of specific steps and operations, in correlation with the conservation state and differentiated by structural elements. The studied icon is from the 18th century and has a special beauty and ornamental complexity, being an exceptional cultural heritage asset, which required urgent preservation and restoration interventions.

As it was in a precollapse state (it could no longer be displayed in the museum), it required immediate preventive consolidation operations and stoppage of the insecto-fungal attack, followed by partial cleaning and varnishing, completion and final consolidation of the wooden support and paint layers, chromatic reintegration of the paintings and regildings, so that finally, before the protection box, it could be repainted, with the restoration of the old patina.

The icon presented superficial and deep gaps in the proportion of 60%. Following the identification of the pictorial materials, gildings and the wooden support, and establishing their conservation state, the adhesion of the pictorial layer to the support was restored by consolidation, involving a gradual system of operations. After removing the adherent dirt, the missing elements and the superficial gaps were filled by repeated grouting, restoring gradually, by specific operations, the physical-structural integrity, and by chromatic reintegration and regilding, the icon was brought as close as possible to its original shape.

OM, SEM-EDX, micro-FTIR methods were used to identify the chemical nature of the composite materials and to determine their preservation state, and CIE L*a*b* colorimetry and visible and UV reflectography were used in the evaluation of the washing, compatibility and synergy studies. Based on the data obtained, the optimal materials and procedures were selected for structural (including wooden support grouting and lacunes filling), chromatic and regilding reintegration, followed by final revarnishing, with or without patination additives.

The consolidation of the pictorial layer, the cleaning of the dirt deposits, the filling of the missing elements and the gaps by grouting, together with the chromatic reintegration and regilding are very delicate, fundamental operations that were carried out in accordance with the unanimously accepted principles of Conservation Science, using compatible materials, with reversibility and legibility in application.

The scientific research undertaken before and during the preservation-restoration operations provided new and important data related to the capitalization and hoarding of the icon as a heritage asset, respectively, data on the evolution of the conservation state, grouped by historical stages. The challenge for the authors was imposed by the alternation of the classic preservation and restoration approaches with the innovative ones, which allowed the integrity of the icon to be preserved without diminishing, through the interventions made, the aesthetic–artistic and the technical–scientific value. Materials similar to the original and modern technologies were used, all of which are descried in the

text. The main purpose was to bring the icon to a shape that would accurately reproduce its function before being introduced into the museum collection and exhibition.

**Author Contributions:** Conceptualization, I.S. and V.V.; methodology, V.V., S.P., A.D. and L.N.; formal analysis, V.V., A.D., S.P. and L.N.; data curation, I.S. and V.V.; writing—original draft preparation, L.N., I.S. and V.V.; writing—review and editing, I.S., V.V. and A.D.; data acquisition, V.V., A.D., S.P. and L.N.; L.N., V.V., A.D., S.P. and I.S. are equally the main authors of this paper. All authors have read and agreed to the published version of the manuscript.

**Funding:** This research received no external funding.

**Conflicts of Interest:** The authors declare no conflict of interest.

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
