# Peer review of "Preservation and Restoration of an Old Wooden Icon with Complex Carved Ornaments, in a Conservation State of Precollapse"

_applsci, doi:10.3390/app12105073_

Round 1

Reviewer 1 Report

This is an interesting paper on the study and conservation of an old wooden icon. However, the manuscript presents many weak points and, in my opinion, it should be better explained and reformulated.

The article presents numerous interesting data, but it is too broad and I think that some paragraphs could be reduced and/or moved in the support information.

The manuscript is very confusing and there are a lot of repetitions and unclear sentences. I would suggest you reorganize the different paragraphs and delete repeated information.

Most of the images might be shifted in the support information.

The captions should be restructured.

The contribution of scientific analyses to the selection of the conservation procedure of the artefact is not clear.

Figure 10: the optical images should have the magnification marker.

Figure 11: SEM micrographs are too small they should be enlarged; furthermore, the operating conditions are not legible.

Table 1: “gravimetric percentages” should be changed to % weight. The table is not easy to read, it should be enlarged.

In my opinion, major revisions are necessary before publication.

Author Response

Thank you for your revision!

Reviewer 2 Report

The manuscript “Preservation and restoration of an old wooden icon with complex carved ornaments, in a conservation state of precollapse” by Liliana Nica and co-authors, aims at the restauration procedures performed on wooden icon with ornaments.

The manuscript is written in a not clear way and with not satisfying the scientific and diagnostic novelties that are required by the journal. The whole paper is focused on the description of the restoration procedures without any discussion of the few diagnostic techniques used. I feel the manuscript out of topics to be published in Applied Sciences.

I suggest to the authors to read the aims and scope of Applied Sciences and to resubmit the paper to a journal that better fit with the restoration procedures.

Author Response

Thank you for your revision!

Reviewer 3 Report

no comments

Author Response

Thank you for your revision!

Round 2

Reviewer 2 Report

Accepted in the present form.